# Swin-HSSAM: A green coffee bean grading method by Swin transformer

**Yujie Jiao**[1,2,3], **Yuqing Zhao**[1,2,3,4], **Aoying Jia**[1,2], **Tianyun Wang**[1,2], **Jiashun Li**[2], **Kaiming Xiang**[1,2], **Hangyu Deng**[1,2], **Maochang He**[1,2], **Rui Jiang**[1,2], **Yue Zhang** [2,3,5]*

**1** Faculty of Mechanical and Electrical Engineering, Yunnan Agriculture University, Kunming, China, **2** Key Laboratory for Crop Production and Smart Agriculture of Yunnan Province, Kunming, China, **3** Yunnan Key Laboratory of Coffee, Kunming, China, **4** Faculty of Transportation Engineering, Kunming University of Science and Technology, Kunming, China, **5** College of Big Data, Yunnan Agricultural University, Kunming, China

* 674406584@qq.com

## Abstract

A novel shifted window (Swin) Transformer coffee bean grading model called Swin-HSSAM has been proposed to address the challenges of accurately classifying green coffee beans and low identification accuracy. This model integrated the Swin Transformer as the backbone network; fused features from the second, third, and fourth stages using the high-level screening-feature pyramid networks module; and incorporated the selective attention module (SAM) for discriminative power enhancement to enhance the feature outputs before classification. Fusion Loss was employed as the loss function. Experimental results on a proprietary coffee bean dataset demonstrate that the Swin-HSSAM model achieved an average grading accuracy of 96.34% for the three grading as well as the nine defect subdivision levels, outperforming the AlexNet, VGG16, ResNet50, MobileNet-v2, Vision Transformer (ViT), and CrossViT models by 3.86%, 2.56%, 0.44%, 4.05%, 5.36%, and 5.40% percentage points, respectively. Evaluations on a public coffee bean dataset revealed that, compared with the aforementioned models, the Swin-HSSAM model improved the average grading accuracy by 1.01%, 0.13%, 4.75%, 0.85%, 0.73%, and 0.27% percentage points, respectively. These results indicate that the Swin-HSSAM model not only achieved high grading accuracy but also exhibited broad applicability, providing a novel solution for the automated grading and identification of green coffee beans.

## 1. Introduction

Coffee, tea, and cocoa, are the three most traded beverages globally. Since the introduction of coffee in China in 1892, Yunnan has since emerged as the country's largest coffee-producing region [1], with coffee production at 113,600 tons, which accounts for 98% of the national output. The size and appearance of green coffee beans directly impact the economic value of coffee products; thus, the grading of

**Data availability statement:** We built our own dataset of tens of thousands of coffee beans, while verifying its generalizability with a copy of the coffee bean data on kaggle .Proof of data availability has been added, links to the dataset have been fixed, and the specified pages can now be viewed. The shared and proprietary datasets used in the text are available and the code and the datasets have been shared to the web page (https://github.com/tony-alice77/Papers-related-to-swin-transformer-improvements/tree/master) [the repository name] is Papers-related-to-swin-transformer-improvements.

**Funding:** The funders had no role in study design, data collection and analysis, decision to publish, or preparation of the manuscript. 1. Yuqing Zhao Yunnan Province Major Science and Technology Special Program Project (202302AE0900200105) https://kjt.yn.gov.cn/; 2. Yuqing Zhao Yunnan Provincial Science and Technology Department Science and Technology Program Agricultural Joint Special Project (202301BD070001-105) https://kjt.yn.gov.cn/ 3. Yuqing Zhao Yunnan Key Laboratory of Coffee (202449CE340030) https://kjt.yn.gov.cn/.

**Competing interests:** The authors have declared that no competing interests exist.

bean size and sorting for defective beans are critical steps in coffee production. As the coffee industry evolved, the international American Specialty Coffee Association (SCA) and domestic coffee associations began developing more detailed professional guidelines for coffee sorting [2,3]. Coffee grading is typically performed using mechanical and manual methods according to increasingly stringent standards, which enhances the complexity of grading. This has limited the ability of mechanical sorting to identify defective beans, while manual sorting, although more effective, is labor-intensive and inefficient [4,5]. Therefore, a method that allows for the rapid, non-destructive sorting of green coffee beans based on the appearance needs to be developed.

Machine learning is now widely employed in tasks that are labor-intensive [6], time-consuming [7], and high-precision [8] in grading. Recent studies on coffee grading have primarily used conventional machine and deep learning techniques. Bazame et al. [9] employed the darknet framework in conjunction with YOLOv3-tiny to identify the ripeness of coffee fruits, achieving an accuracy of 83%. Chou et al. [10] developed a deep learning-based defective bean inspection scheme, along with an automatic data augmentation method using a generative adversarial networks structure to enhance the proposed scheme, which demonstrated 80% accuracy in identifying defective beans. Chang et al. [5] introduced a novel deep learning approach for detecting eight types of defective coffee beans, obtaining an accuracy rate of 95.2%. Akbar et al. [11] employed color histograms and local binary patterns to extract features from green coffee beans, subsequently employing random forest and k-nearest neighbors algorithms for grading, achieving accuracies of 87.87% and 80.47%, respectively. Zhao et al. [12] used machine vision technology to extract three types of coffee bean features and employed a support vector machine for defect grading, achieving an accuracy of 84.9%.

Deep learning is a specialized subset of machine learning. The prominence of transformer models in natural language processing is well established within the deep learning community [13]. The unique attention mechanism of the transformer is utilized to capture non-local dependencies to achieve a broader receptive field [14]. Because of its robust performance, its applications span a wide array of fields, including speech recognition [15], object detection [16], video understanding [17], and multimodal learning [18]. Enhanced image processing capabilities have been added to transformer architectures, including the Vision Transformer (ViT) [19] and shifted window (Swin) Transformer [20]. The exceptional performance of ViT and Swin Transformer [21] in image recognition tasks demonstrates the potential of transformers in visual applications. Wang et al. [22] developed a SwinGD algorithm based on the Swin Transformer for identifying grape clusters, which achieved a mean average precision (mAP) of 94% at an Intersection of Union of 0.5. Si et al. [23] introduced a dual-branch model, DBCoST, which integrates convolutional neural network (CNN) and Swin Transformer technologies and includes the feature fusion module, which consists of a residual module and an enhanced Squeeze-and-Excitation network. The model boasts a recognition accuracy of 97.32% for diseased apple leaves.

This paper introduces a novel method of green coffee bean grading named Swin-HSSAM, which uses the Swin Transformer network for feature extraction. The

model was trained using the newly proposed Fusion Loss, and the trained features were further enhanced through an enhanced hierarchical scale-based feature pyramid network (HS-FPN) and selective attention module (SAM) structures. This approach facilitated the grading and identification of green coffee beans while simultaneously sorting out defective ones. The new model not only improved the detection accuracy but expedited detection, providing a theoretical foundation for the development of future green coffee bean grading systems.

## 2. Materials and methods

### 2.1 Dataset and experimental environment

This study evaluated green Arabica coffee bean samples collected by Changmu Coffee Company, Pu'er City, Yunnan, China, and manually sorted by globally certified coffee quality appraisers. The samples totaled 10,378 beans. The classification standards used are as outlined in Table 1 [3].

However, the classification of defective beans in this grading standard is rather vague and lacks detailed categorization. Therefore, to enhance the training outcomes and enrich the dataset, we have supplemented it with the types of defective beans delineated in the guidelines recommended by the SCA. The categories of defective beans as defined in the guidelines are shown in Fig 1.

The images were categorized into four types, comprising three classes of normal beans with a total of 7,894 images, and 2,484 images of substandard beans. Specifically, the normal beans include as follows: 2,460 images of first-grade beans, 2,601 images of second-grade beans, and 2,833 images of third-grade beans. The substandard beans include 9 types:black, broken, cherry pods, fungus damage, husk, immature, parchment, shells and sour. The distribution of data is detailed in Table 2. A graphical representation of the dataset is illustrated in Fig 2.

**Table 1. Grading Standards for Green Coffee Beans.**

| Grade | Description |
|---|---|
| First Grade | Round and clean surface, no blemishes visible, smooth, good luster, size >7.1 mm |
| Second Grade | Round and clean surface, slight blemishes visible, smooth, good luster, size between 6.7–7.1 mm |
| Third Grade | Round and clean surface, blemishes visible, smooth, good luster, size between 5.0–6.7 mm |
| Defects | Shape irregular, surface rough, luster not good |

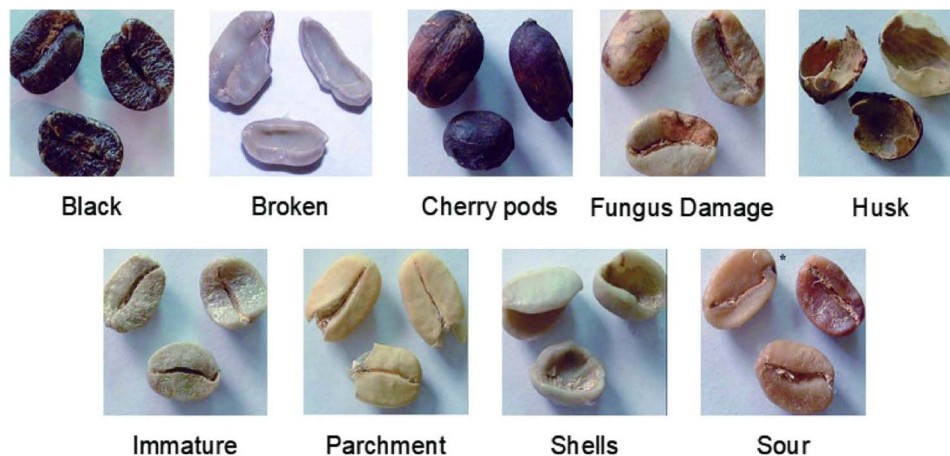

**Fig 1. Different types of defective coffee beans.**

**Table 2. Number and resolution of images.**

| Grade | Training Dataset | Test Dataset | Total |
|---|---|---|---|
| First Grade | 1968 | 492 | 2460 |
| Second Grade | 2081 | 520 | 2601 |
| Third Grade | 2267 | 566 | 2833 |
| Defects | 1988 | 496 | 2484 |

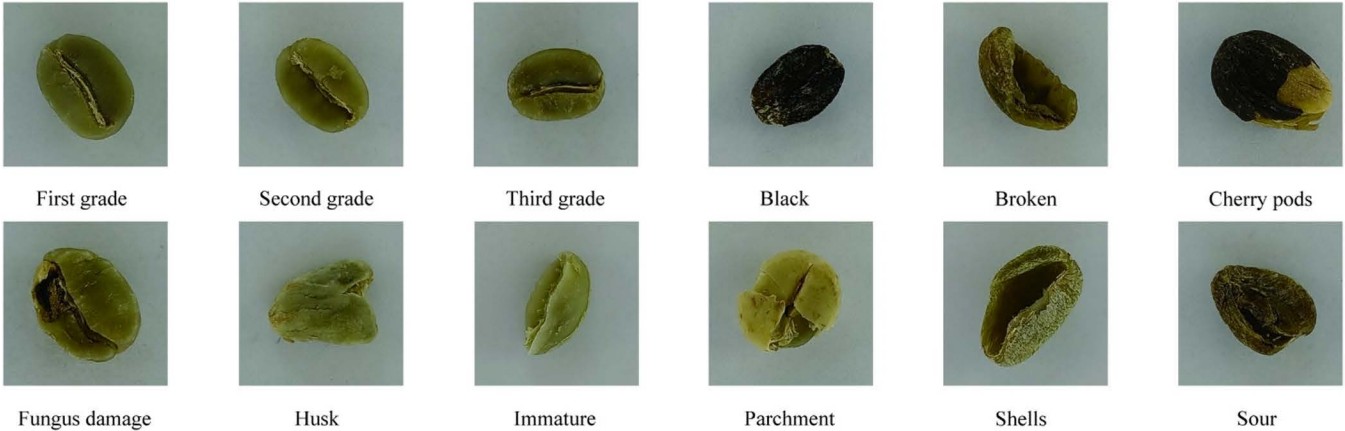

**Fig 2. Different grades of green coffee beans.**

Images were captured using a Huawei Enjoy 20 pro from a height of 140mm. The sample images were first analyzed in their original form and then converted to grayscale. Gaussian filtering was applied to remove noise interference, followed by gamma transformation to enhance the overall details and increase contrast. Edges of the image were sharpened using the Canny operator. Based on differences in grayscale characteristics, a fixed threshold between 175 and 255 was selected to convert the image into binary format, effectively segregating the target from the background. The images were then inverted to enhance the feature areas. Three rounds of opening and closing operations were performed to fill gaps and eliminate edge burrs. Contour tracing code was used to identify the largest contour of the raw coffee bean image, capturing the smallest bounding rectangle of the raw image. To minimize errors, the width and height of the bounding rectangle were each expanded by 10 pixels, and the image were cropped to a size of 224×224 pixels [13], as shown in Fig 3 The normal and defective beans across the three grades were then divided into a training set and a test set in an 8:2 ratio, which consisted of 8,304 images for training and 2,074 images for testing.

The experimental platform for the study was configured as follows: The computer used had an Intel Xeon Gold 6230R processor and was equipped with an NVIDIA RTX A6000 graphics card. All models were constructed using the Pytorch 1.13.1 deep learning framework on the Jupyter platform, with Python 3.9 as the programming language. Model training employed the AdamW optimizer, with the initial learning rate set at 0.00005. A StepLR scheduler was used for the learning rate adjustment, which halved the rate every 50 epochs.

## 2.2 Swin transformer

The Swin Transformer is a deep learning model based on the foundational principles of the Transformer. Compared with the ViT, the Swin Transformer is more efficient and accurate. As shown in Fig 4a and 4b, the Swin Transformer reduced computational complexity by partitioning the feature map into smaller windows, unlike the ViT structure. It constructed a hierarchical representation by starting with small-sized patches and progressively merging adjacent patches layer by layer. However, while

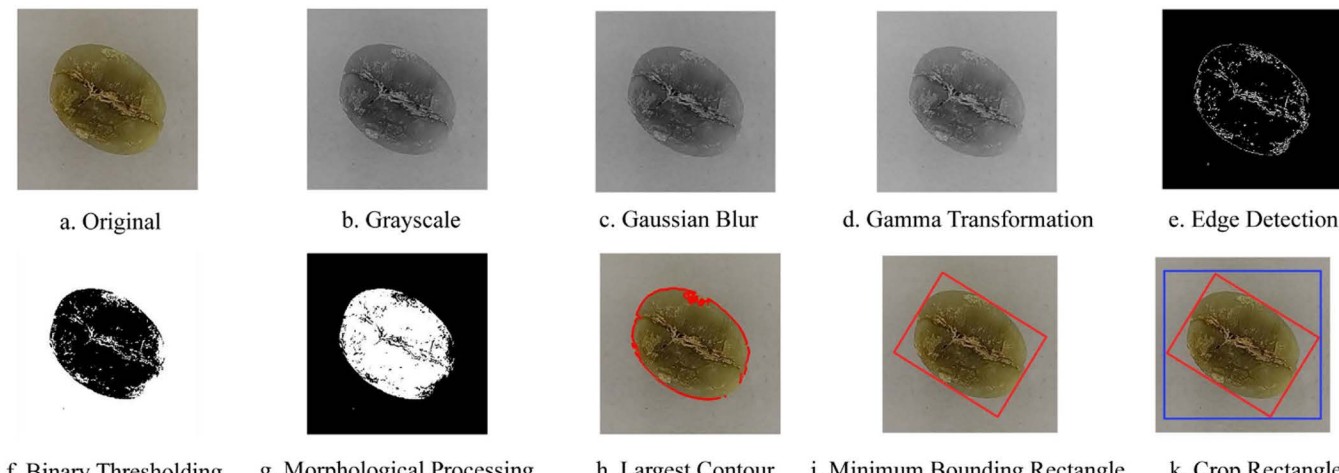

Fig 3. **Image data preprocessing of Arabica green coffee beans.**

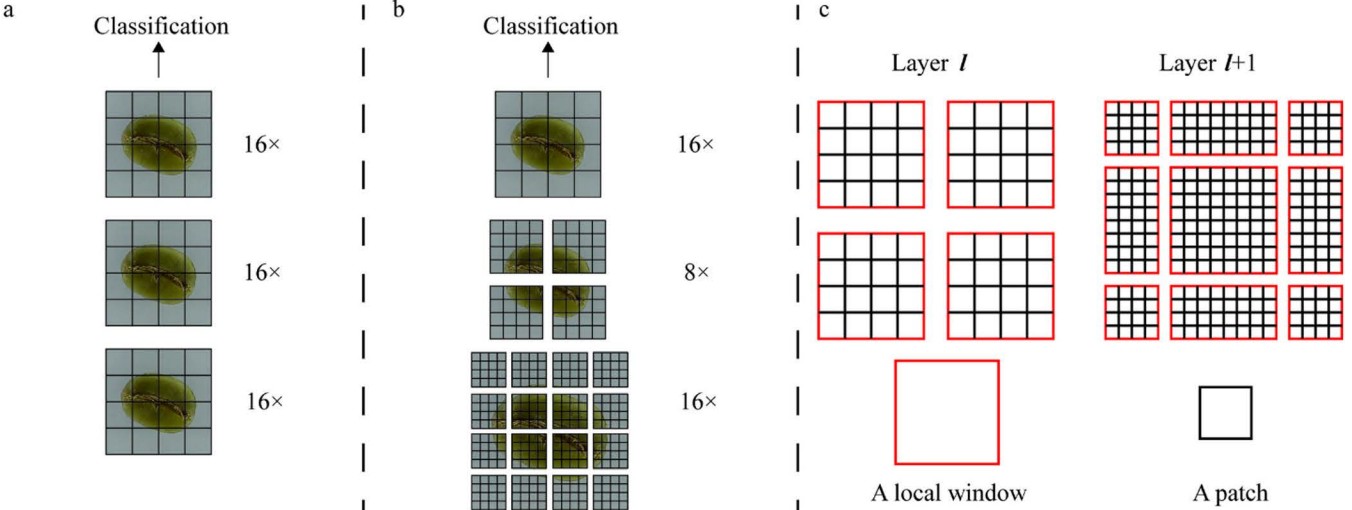

Fig 4. **Feature maps structure. a** The feature maps of previous ViT; **b** the hierarchical feature maps approach; **c** the shifted window approach for computing self-attention.

it reduced the computational load, this method also impeded the exchange of information between windows [24]. To address this issue, the Swin Transformer employed a technique of shifting the divided windows, as shown in Fig 4c. This shift facilitated information exchange between windows that would normally not communicate without increasing computational complexity.

Fig 5 shows the fundamental architecture of the Swin Transformer. Using the patch splitting module, the input RGB image was initially divided into the smallest unit tokens, i.e., non-overlapping patches. This module is composed of two components, namely, Patch Partition and Linear Embedding. Subsequently, through overlapping Swin Transformer blocks, the receptive field was expanded, model capacity was enhanced, and cross-window information fusion was achieved. Ultimately, this process completed the task of feature representation learning, which was used to generate patch tokens for creating hierarchical feature representations.

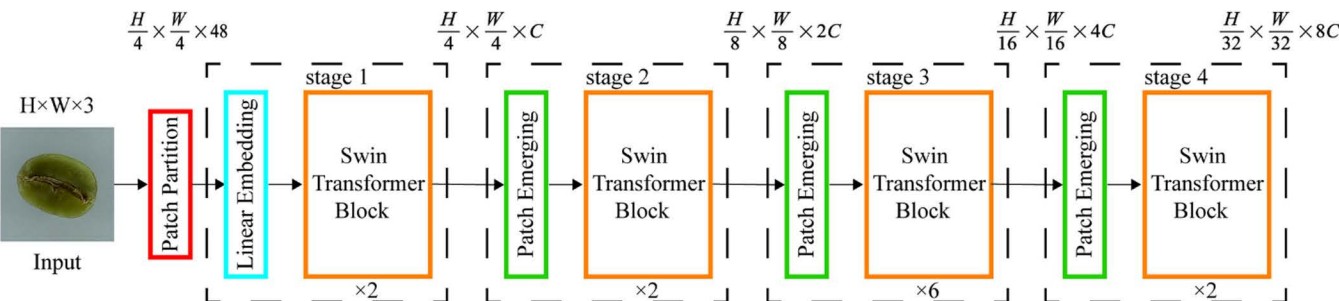

**Fig 5. Shifted window Transformer architecture.**

## 2.3 Swin Transformer block

The Swin Transformer Block was constructed based on shifted windows. As illustrated, a Swin Transformer Block typically consists of an even number of blocks forming one stage, Stages consisting of different numbers of Swin Transformer Blocks can form different Swin Transformer variants,with the Swin-T(a variant structure of the Swin Transformer) model generally comprising 2, 2, 6, and 2 blocks per stage. Each block contains four components: a layer norm (LN) layer, multi-head self-attention (MSA) module, residual connection, and two-layer multilayer perceptron (MLP). Odd-numbered blocks utilized the window-based MSA(W-MSA) module [25], while even-numbered blocks employ the shifted window-based MSA (SW-MSA) module. Following this organizational scheme, feature maps were calculated using the formulas below:

$$\hat{z}^l = W-MSA(LM(z^{l-1}) + z^{l-1} \tag{1}$$

$$z^l = MLP(LN(\hat{z}^l) + \hat{z}^l \tag{2}$$

$$\hat{z}^{l+1} = SW-MSA(LN(z^l) + z^l \tag{3}$$

$$z^{l+1} = MLP(LN(\hat{z}^{l+1})) + \hat{z}^{l+1} \tag{4}$$

In the block structure (Fig 6), $\hat{z}^l$ and $\hat{z}^{l+1}$ illustrate the outputs of the $l$th W-MSA module and the subsequent SW-MSA module, respectively. The symbol $z^l$ indicates the output of the $l$th multilayer perceptron (MLP) module.

## 2.4 High-level screening-feature fusion pyramid

Identifying objects in computer vision remains difficult due to their scale variation. Feature pyramids, which are built on image pyramids, form a fundamental solution to this challenge [26]. They adapt to object scale changes by switching between different levels within the pyramid, while the scale of the pyramid itself remains unchanged. The advent of deep learning has enabled the integration of pyramid structures into CNNs to establish multi-scale representations of images. Lin et al. [27] introduced a novel architecture known as FPN, which combines features from different layers to form feature maps of varying scales within the feature pyramid. Using FPN, a greater integration of shallow feature map information is achieved, enhancing small object detection accuracy and providing more robust semantic information.

Chen et al. [28] developed an HS-FPN(High-level Screening-feature Fusion Pyramid) for merging multi-scale features, especially for finer details, such as leukocytes. This enhancement enabled the model to capture a more comprehensive

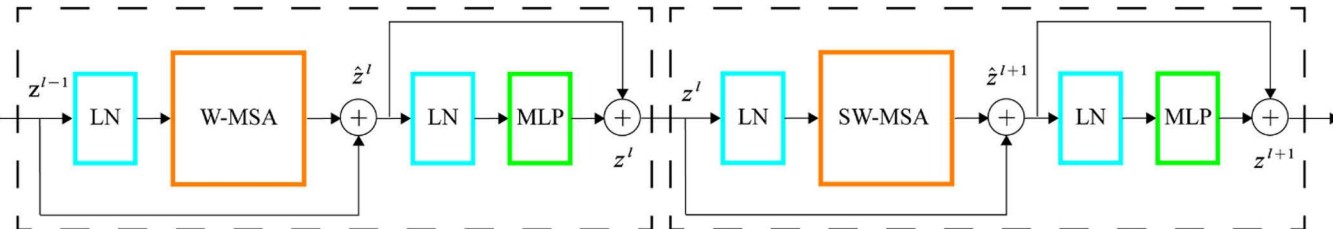

**Fig 6. Swin Transformer block.** LN, layer norm layer; W-MSA, window-based multi-head self-attention module; MLP, multilayer perceptron. $Z_l$, outputs of the $l$th W-MSA module; $\hat{z}^{l+1}$, shifted window-based MSA module; $Z_l$ output of the $l$th MLP module.

representation of leukocyte features. The structure of the HS-FPN, as depicted in Fig 7, consists of two primary components, namely, a feature selection module and a feature fusion module.

Initially, feature maps of varying scales undergo a selection process within the feature selection module. Subsequently, through the selective feature fusion (SFF) mechanism, high- and low-level information are synergistically integrated within these feature maps. This fusion enhances the model's detection capabilities by generating features rich in semantic content, especially useful for detecting small-scale features. Fig 7 illustrates the fundamental architecture of the HS-FPN.

## 2.5 SAM for discriminative power enhancement

SAM(selective attention module) comprises three primary components: the control depthwise separable convolution (CDSC) module [29], a fully connected layer, and an exponential channel component. The architecture is illustrated Fig 8 provided.

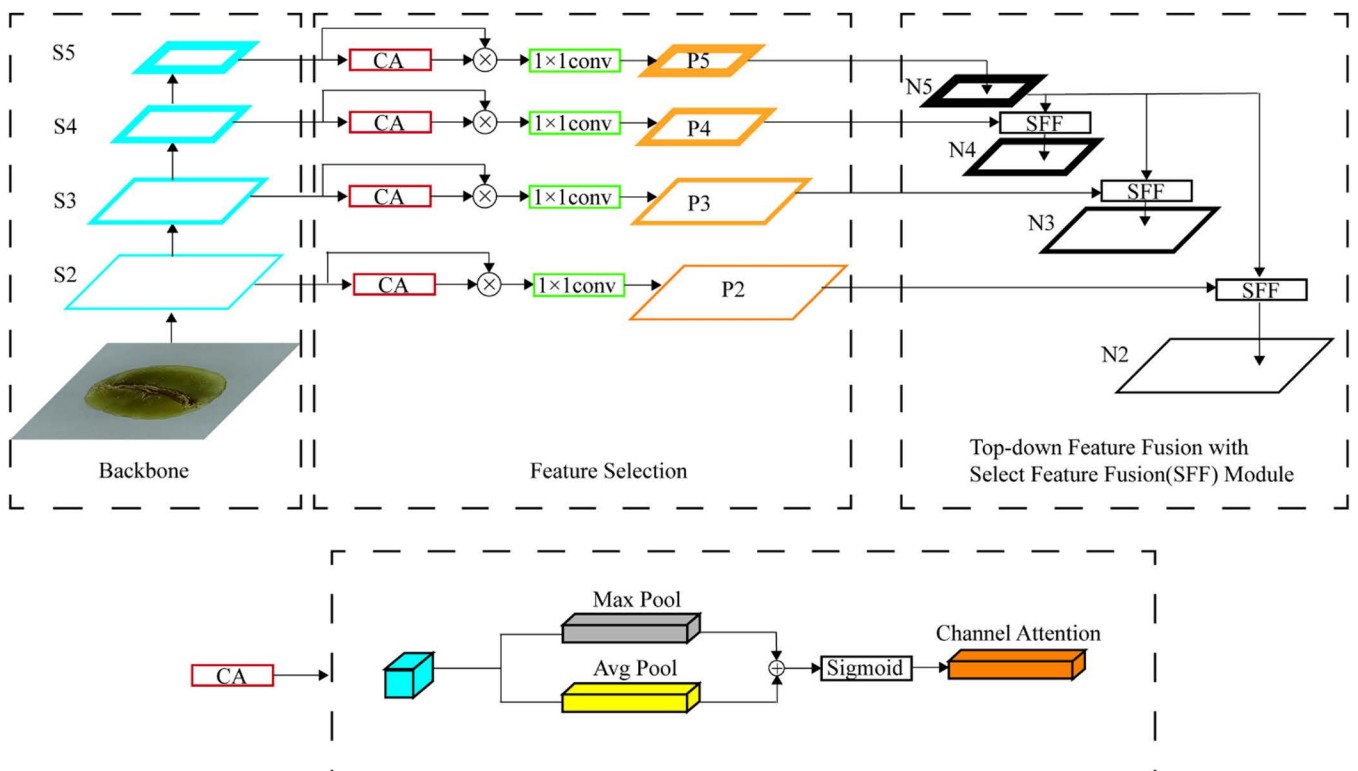

**Fig 7. Architecture of the hierarchical scale-based feature pyramid network.**

The CDSC module features depthwise and pointwise convolution, as depicted. The number of output feature maps in each depthwise convolution equals the number of input channels. Using depthwise convolution instead of standard convolution saves computational resources, as evident in the given equations [30]. The fundamental architecture of the SAM and CDSC is depicted in Figs 8 and 9, respectively.

$$\begin{cases} \cos t_{\text{Conv}} = h * w * \text{Cin} * \text{Cout} * K * K \\ \cos t_{\text{DSC}} = h * w * \text{Cin} * (K^2 + \text{Cout}) \end{cases} \tag{5}$$

In Equation 5, Cin denotes the number of channels in the input feature map; Cout represents the number of channels in the output feature map; and K indicates the size of the convolution kernel.

Compared with traditional convolution, CDSC reduces the computational cost by decreasing the number of parameters, achieving a reduction factor of 1/K2. The results were then processed through the Gelu function and normalized before being fed into the pointwise convolution. This pointwise convolution adeptly combines feature maps to generate new ones. The expression of the Gelu function, as shown in the formula, implements a linear transformation that zeroes out non-essential neural activations while augmenting the network's capability to represent intricate data distributions and patterns.

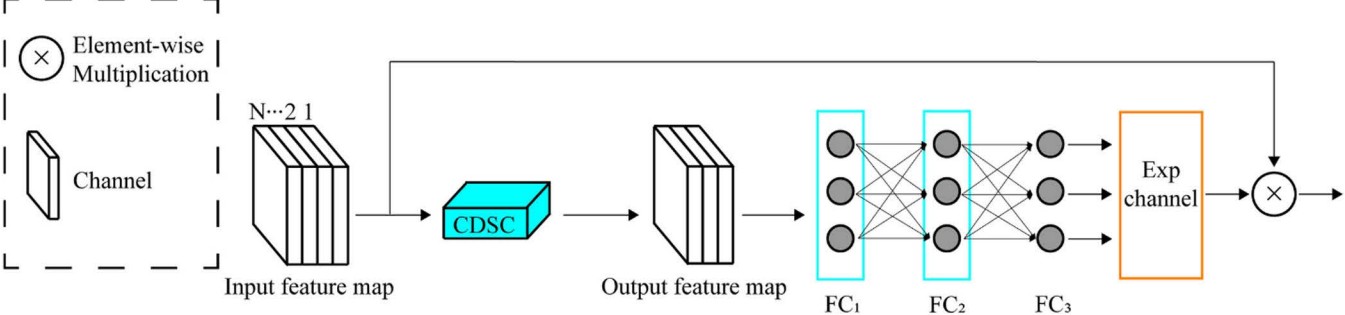

**Fig 8. SAM discriminability enhancement module structure diagram.**

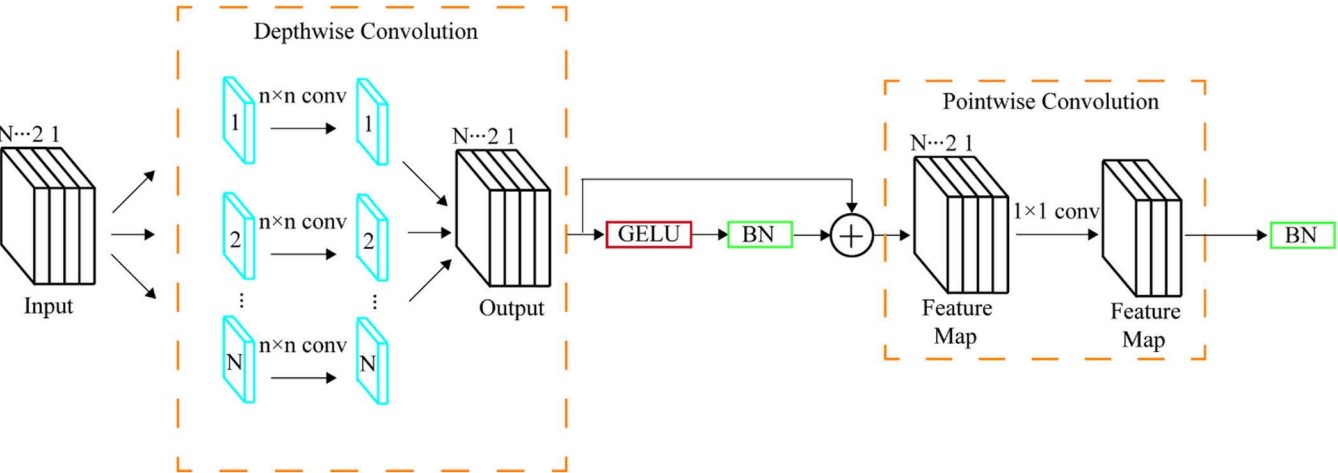

**Fig 9. Detailed structure diagram of the control depthwise separable convolution module.**

$$GELU(x) = 0.5x \left( 1 + \tan h \left( \sqrt{\frac{2}{\pi}(x + 0.44715x^3)} \right) \right)$$

(6)

## 2.6 Fusion loss

Fusion Loss ingeniously combines Focal Loss and Cross-Entropy Loss. For each sample, it calculates the average of Focal Loss and Cross-Entropy Loss, then assigns weights (weight_ce and weight_focal) respectively to the average value of them. The sum of these weighted values yields the composite loss value. Fusion Loss notably improves model performance by reducing loss while maintaining recognition accuracy [31].

Fusion Loss penalizes incorrectly predicted samples, enhancing the model's accuracy in learning target categories. This helps address issues of class imbalance by swiftly focusing on samples that are difficult to distinguish. The formula for Fusion Loss is as follows:

$$FL(p_i) = -w_f a_t (1 - p_t)^y \log(p_t) - w_c \sum_{i=1}^{n} q_i \log(p_i)$$

(7)

where $p_t$ represents the predicted probability; $a_t$ is the adjustment factor balancing the importance of positive and negative samples; $y$ is the focusing parameter that diminishes the weight of easily classified samples; $q_i$ is a binary variable that is 1 when the condition is satisfied otherwise it is 0 and $w_f$ and $w_c$ are the parameter values for weight_focal and weight_ce, respectively.

## 2.7 Swin-HSSAM

Using the Swin-T model as the foundational architecture significantly reduces computational demands during model training. The initial input size for green coffee beans as 224×224 pixels, which, following the patch partition operation, was resized to 56×56. After undergoing transformations through four stages, the output dimension was further reduced to 7×7. Each stage producds feature maps at four distinct scales: S2, S3 S4 and S5, with the spatial size of each feature map diminishing with increasing depth. The feature maps S3, S4, and S5 were transformed into P3, P4, and P5 through a channel attention mechanism and a 1×1 conversion. Outputs from the feature selection phase were directed into the feature fusion section, where each feature map was upsampled to align with spatial resolutions. In the feature fusion section, a top-down fusion strategy was implemented, culminating in the output N3 from the SFF feature fusion module. This output N3 was subsequently enhanced by the SAM module and linked to a fully connected layer for classification. The model employed Fusion Loss as its loss function.

Shallower features can more effectively identify smaller objects. Therefore, the Swin-HSSAM model enhances the Swin-T one by incorporating an HS-FPN structure, which effectively integrates shallow features into the final output. Adding the SAM structure before classification further enhances the learning capabilities for target features and local feature information across each channel. The receptive field exponentially expands as the layers deepen, significantly improving the model's accuracy in identifying small targets like coffee beans. Fig 10 is a schematic diagram of the specific structure of Swin-HSSAM.

## 3. Experimental results and analysis

### 3.1 Evaluation criteria

The performance of various models were evaluated using metrics such as mAP, accuracy, F1-score, and speed (Frames Per Second,FPS). True positive, true negative, false positive (FP),false negative, recall (R), and precision (P) were used

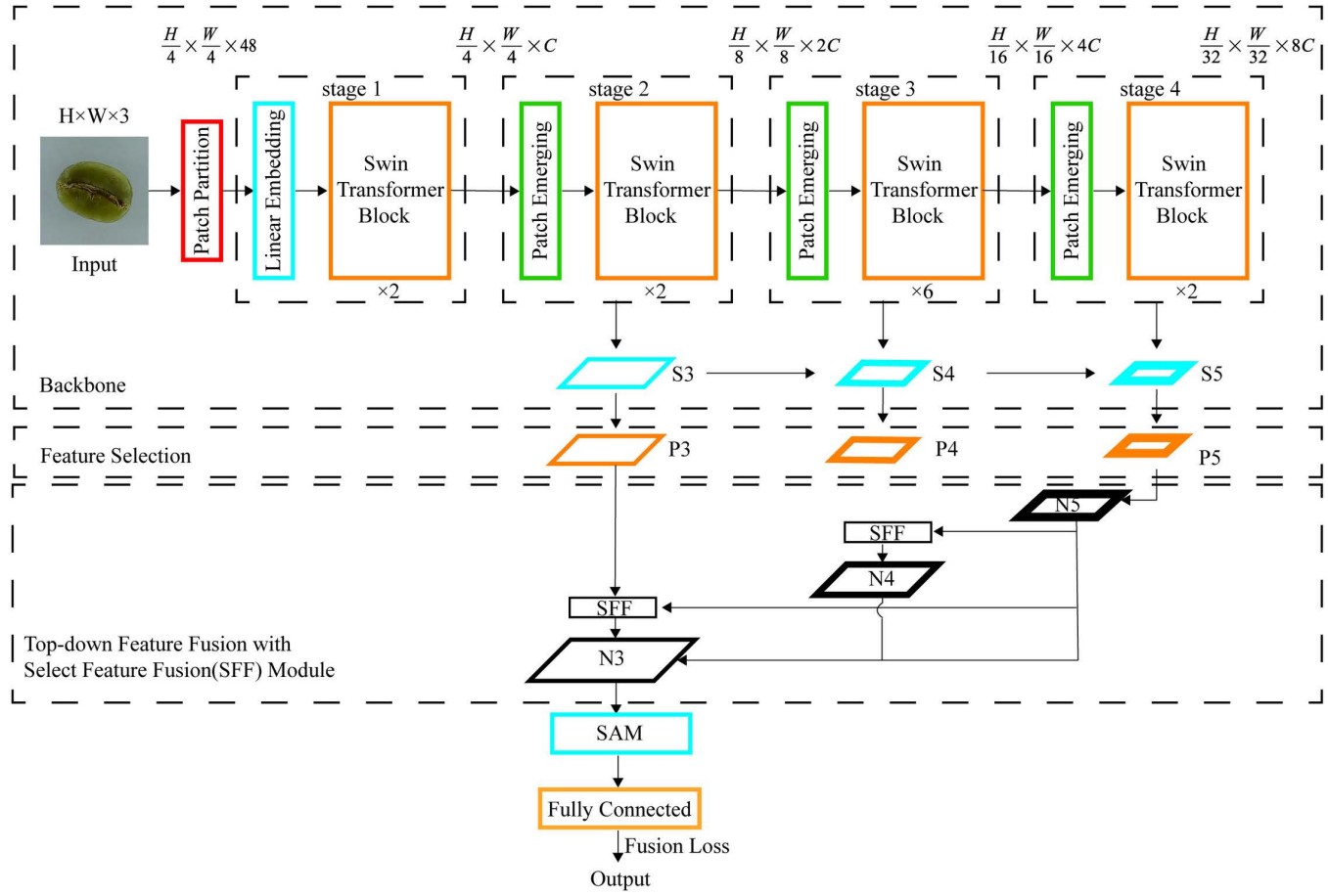

**Fig 10. Swin-HSSAM architecture.**

to define these five criteria in (Eq. 8–10). By comparing these metrics, the models were assessed against each other, and losses were calculated using Fusion Loss.

$$mAP = \frac{1}{N} \sum_{i=1}^{N} AP \tag{8}$$

$$Accuracy = \frac{TP + TN}{TP + TN + FP + FN} \tag{9}$$

$$F1 = \frac{2 \times P \times R}{P + R} \tag{10}$$

## 3.2 Swin-HSSAM modeling tests

Compared with the baseline model Swin-T, the mAP metric of the Swin-HSSAM model was better by 1.31 percentage points. As shown in Table 3, both average accuracy and F1 values improved by 3.5 percentage points. The incorporation of HS-FPN into the baseline model enables the capture of more detailed features from green coffee beans, while the feature fusion achieved by SAM enhances semantic content, leading to improved detection accuracy, Furthermore, the

**Table 3. Complexities between the Swin-HSSAM and Swin-T models.**

| Model | mAP | Average accuracy (%) | F1 score (%) |
|---|---|---|---|
| Swin-T | 97.20 | 92.84 | 92.85 |
| **Swin-HSSAM** | 98.51 | 96.34 | 96.35 |

increase in the F1 score indicates that the Swin-HSSAM model reduces both false positives and missed positives compared to the baseline. Table 4 depicts the performance details of the Swin-HSSAM model on nine defective bean classifications. As shown in Table 4, the detection accuracy for six types of defective beans exceeded 95%, except for Parchment, Immature, and Fungus damage, which achieved accuracy above 90%. However, Parchment had a lower detection accuracy of 86.5%, likely due to its limited sample size and the minimal visual distinction from normal beans.

Fig 11 and 12 illustrate the training and validation loss curves, highlighting the convergence behavior of the detection model. As shown, the Swin-HSSAM model achieves significantly lower loss and faster convergence compared to the baseline model. This improvement demonstrates that the introduction of Fusion Loss addresses the imbalance in defective green coffee beans and the challenging classification of normal beans, thereby enhancing model accuracy.

### 3.3 Impact of different Fusion Loss weights on model performance

The combined Fusion Loss and Swin-T model yielded excellent experimental results and achieved remarkable identification accuracy. This section explores the impact on model performance of employing different fusion weights for combining Focal Loss and Cross-Entropy Loss. Nine distinct weight combinations were tested, while all other variables remained constant. The results presented in Table 5 indicate that the best performance was achieved when the weights for Cross-Entropy Loss and Focal Loss were set to 0.7 and 0.3, respectively. This configuration resulted in the highest mAP, Accuracy, and F1 scores, which were 97.82%, 93.74%, and 93.74%, respectively. The weights of Cross-Entropy Loss and Focal Loss are set to 0.7 and 0.3, respectively, indicating that Cross-Entropy Loss plays a dominant role, while Focal Loss assumes a secondary position. This configuration directs the training process to prioritize minimizing Cross-Entropy Loss, which emphasizes reducing the discrepancy between the predicted and true probability distributions. Simultaneously, it ensures attention to hard-to-classify samples, thereby enhancing the model's overall performance.

### 3.4 Impact of different HS-FPN structure on model performance

This section describes the comparative experiment performed to adjust the standard HS-FPN structure to verify the most effective stages of information fusion for the grading of green coffee beans. The experimental results shown in Table 6 indicate that the model structure combining S3、 S4 and S5 chieved the highest mAP, accuracy, and F1 scores

**Table 4. Complexities between the different defective types coffee beans (Swin-HSSAM).**

| Different types of defective coffee beans | mAP | Average accuracy (%) | F1 (%) |
|---|---|---|---|
| Black | 98.50 | 98.48 | 98.48 |
| Broken | 99.11 | 99.03 | 99.06 |
| Cherry pods | 99.42 | 99.37 | 99.38 |
| Fungus damage | 93.35 | 91.04 | 91.05 |
| Husk | 99.50 | 99.10 | 99.11 |
| Immature | 92.80 | 92.29 | 92.30 |
| Parchment | 88.86 | 86.50 | 96.52 |
| Shells | 98.00 | 93.81 | 93.83 |
| Sour | 95.84 | 94.62 | 94.63 |

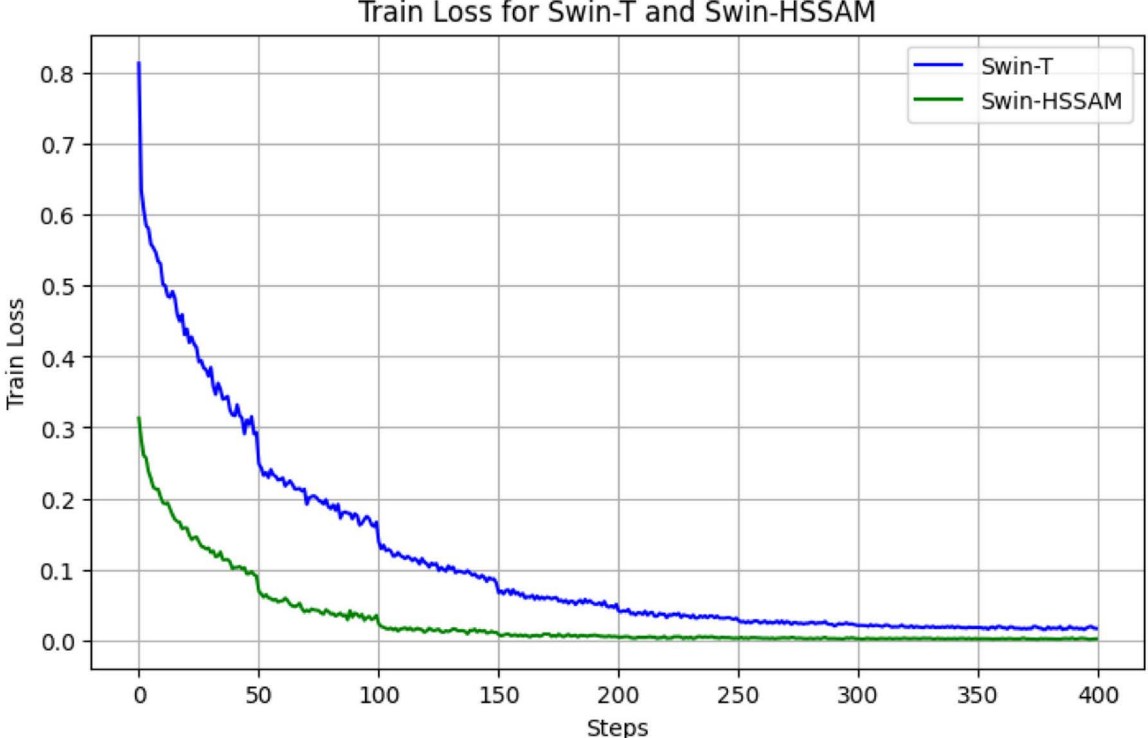

**Fig 11. Loss change curve of the Swin-HSSAM and Swin-T models(Train).**

at 98.51%, 96.34%, and 96.35%, respectively. Graphs illustrating the loss changes for different HS-FPN structures are presented in Fig 13.

The results indicate that deeper information layers are more advantageous for recognizing small targets such as coffee beans. The blank areas in global information likely impacted the identification of coffee bean grades, thereby decreasing accuracy. Consequently, this experiment was chosen to employ an HS-FPN structure that fuses information from S3 S4 and S5.

### 3.5 Ablation study of the Swin-HSSAM model

An ablation study was performed to evaluate the influence of the HS-FPN module, SAM module, and Fusion Loss on the model's stability and accuracy. Table 7 indicates that using only the HS-FPN with the Swin-T grading model led to a 0.31 percentage-point increase in mAP and a 0.65 percentage-point improvement in accuracy. The Fusion Loss model saw a 0.62 percentage-point increase in mAP and a 0.90 percentage-point increase in accuracy. The SAM model increased the mAP by 0.65 percentage points and accuracy by 1.67 percentage points. The combined HS-FPN and Fusion Loss model resulted in a 0.65 percentage-point increase in mAP and a 1.00 percentage-point increase in accuracy. The combined HS-FPN and SAM model yielded a 1.17 percentage-point increase in mAP and a 2.30 percentage-point improvement in accuracy. The combined SAM and Fusion Loss model enhanced the mAP by 0.87 percentage point and accuracy by 2.59 percentage points. Lastly, the combined HS-FPN, SAM, and Fusion Loss model increased the mAP by 1.31 percentage points and accuracy by 3.50 percentage points. These ablation studies demonstrate that the introduction of the HS-FPN, SAM, and Fusion Loss modules simultaneously enhanced the model's accuracy and stability.

                                                                     

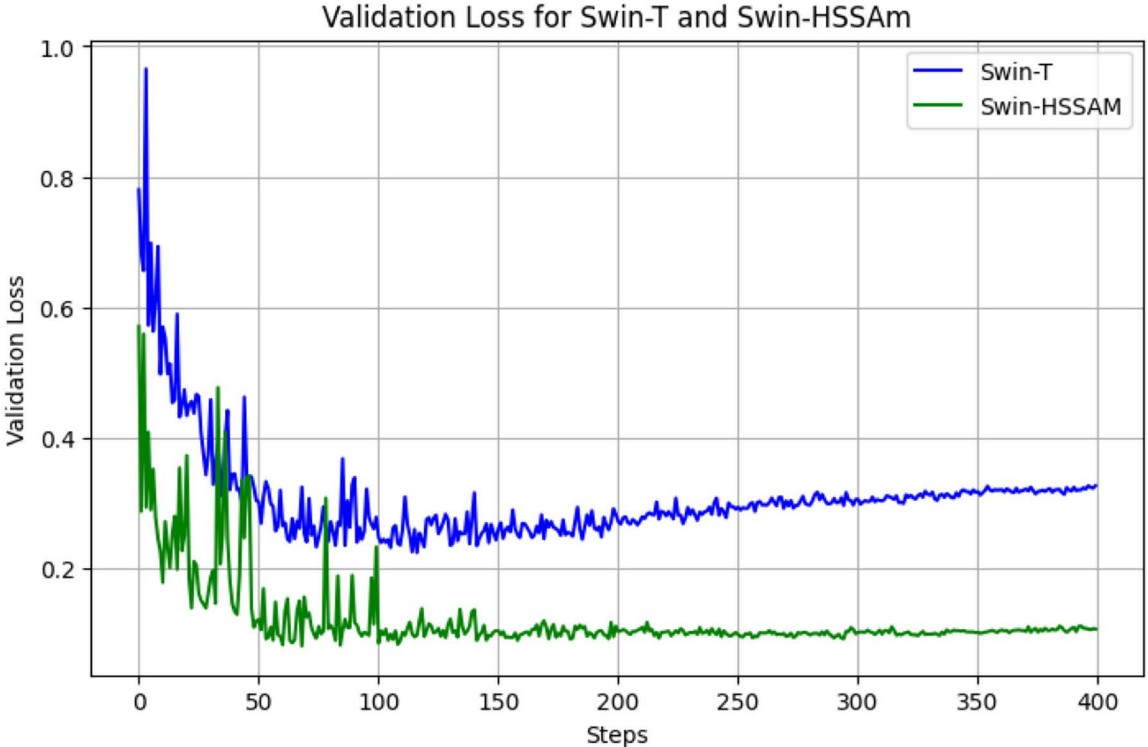

**Fig 12. Loss change curve of the Swin-HSSAM and Swin-T models(Validation).**

**Table 5. Comparison of complexity between different loss.**

| Weight ce | Weight focal | mAP (%) | Average Accuracy (%) | F1 (%) |
|---|---|---|---|---|
| 0.1 | 0.9 | 97.58 | 91.94 | 91.96 |
| 0.2 | 0.8 | 97.80 | 92.66 | 92.68 |
| 0.3 | 0.7 | 97.75 | 92.76 | 92.77 |
| 0.4 | 0.6 | 97.52 | 91.84 | 91.86 |
| 0.5 | 0.5 | 97.49 | 92.14 | 92.14 |
| 0.6 | 0.4 | 97.74 | 92.61 | 92.62 |
| **0.7** | **0.3** | **97.82** | **93.74** | **93.74** |
| 0.8 | 0.2 | 97.58 | 92.42 | 92.44 |
| 0.9 | 0.1 | 97.81 | 92.14 | 92.15 |

mAP, mean average precision.

## 3.6 Proposed method versus other models

To evaluate the performance of the proposed Swin-HSSAM method, a series of were conducted. This study compared the developed method with AlexNet [32], VGG16 [33], ResNet50 [34,35], MobileNet-v2 [36], Vision Transformer (ViT) [19], and CrossViT [37]. The Swin-HSSAM achieved an accuracy of 96.34% and an mAP of 98.51%, surpassing AlexNet, VGG16, ResNet50, MobileNet-v2, Vision Transformer (ViT), and CrossViT models by 3.86%, 2.56%, 0.44%, 4.05%, 5.36%, and 5.40% in accuracy, respectively, and 1.70%, 0.47%, 0.58%, 0.92%, 2.31, and 1.88% in mAP, respectively. The variations in loss and accuracy are shown in Figs 14 and 15. Table 8 presents a comparison of the proposed method with other models. The F1 score, which combines precision and recall,

**Table 6. Comparison of complexity between different HS-FPN structure.**

| Stage number of HS-FPN module fusion | | | | mAP (%) | Average Accuracy (%) | F1 (%) |
|---|---|---|---|---|---|---|
| 2 | 3 | 4 | 5 | | | |
| √ | √ | √ | √ | 98.46 | 96.15 | 96.14 |
| × | √ | √ | √ | **98.51** | **96.34** | **96.35** |
| √ | × | √ | √ | 98.36 | 95.52 | 95.53 |
| √ | √ | × | √ | 98.42 | 95.90 | 95.90 |
| √ | √ | √ | × | 98.38 | 95.71 | 95.72 |

HS-FPN, hierarchical scale-based feature pyramid network; mAP, mean average precision.

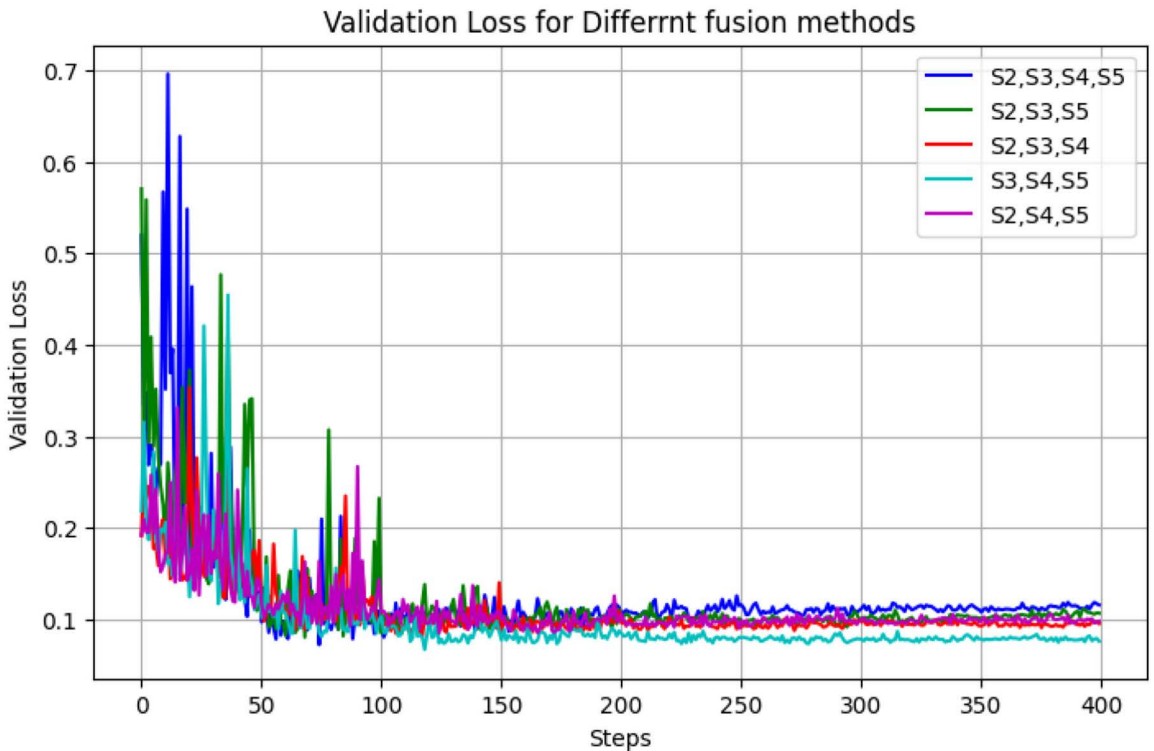

**Fig 13. Loss change curve of different hierarchical scale-based feature pyramid network structure.**

ranges from 0 to 1, with 1 representing optimal model performance and 0 representing the poorest. The F1 scores for first grade and defective beans are higher than those for second- and third-grade beans. This discrepancy may be attributed to the minimal size difference between second- and third-grade beans, causing more frequent misjudgments and confusion by the program, thereby reducing the F1 scores. Comparing the Swin-HSSAM model with more complex deep learning models, such as ViT and Cross-ViT, revealed that the parameters and flops did not excessively increase to ensure the recognition speed while maintaining a high accuracy and F1 value. Fig 14 demonstrates that the Swin-HSSAM model achieves faster convergence and lower loss compared to other models. Similarly, Fig 15 shows that Swin-HSSAM attains the highest accuracy among all models evaluated

The experimental results demonstrate that the proposed Swin-HSSAM model not only performed well but also maintained a rapid identification speed compared with other models.

**Table 7. Ablation study of Swin-HSSAM.**

| Model | Factor | | | mAP | Average Accuracy (%) | F1 score (%) |
|---|---|---|---|---|---|---|
| | **HS-FPN** | **Fusion Loss** | **SAM** | | | |
| Swin-T | × | × | × | 97.20 | 92.84 | 92.85 |
| | √ | × | × | 97.51 | 93.49 | 93.50 |
| | × | √ | × | 97.82 | 93.74 | 93.75 |
| | × | × | √ | 97.85 | 94.51 | 94.52 |
| | √ | √ | × | 97.85 | 93.84 | 93.81 |
| | × | √ | √ | 98.07 | 95.43 | 95.44 |
| | √ | × | √ | 98.37 | 95.14 | 95.14 |
| **Swin-HSSAM** | √ | √ | √ | 98.51 | 96.34 | 96.35 |

HS-FPN, hierarchical scale-based feature pyramid network; SAM, selective attention module; mAP, mean average precision.

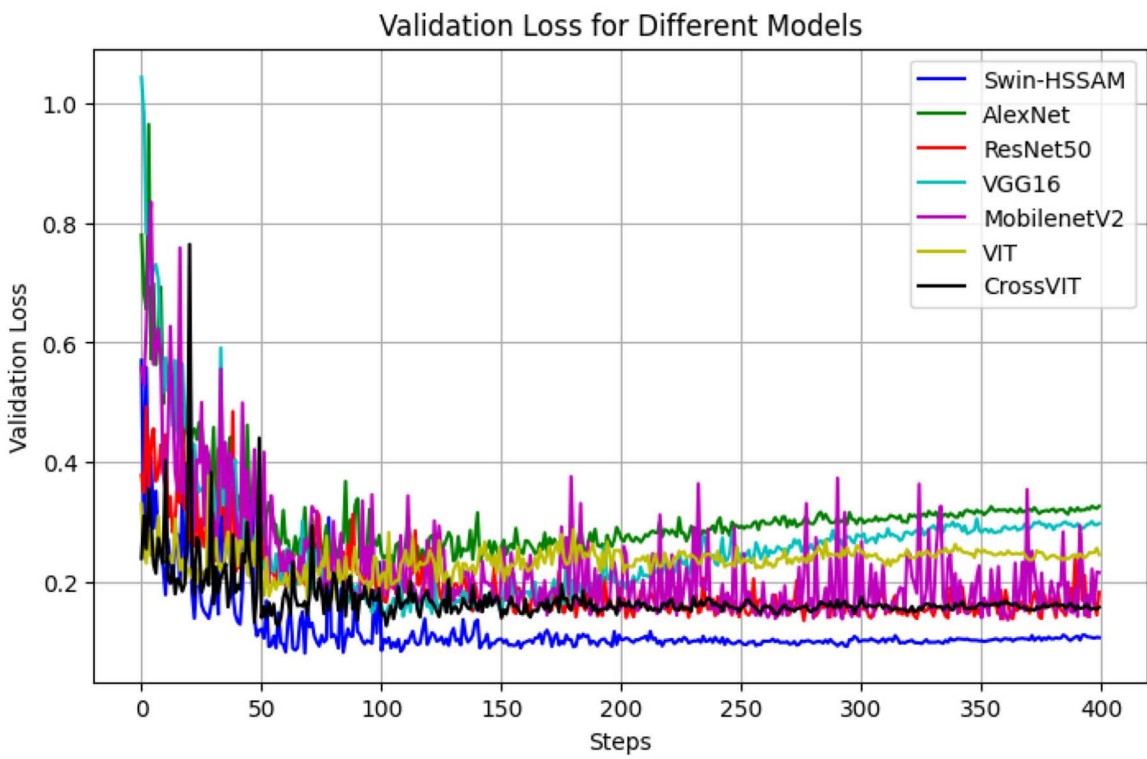

**Fig 14. Loss change curve of different model.**

### 3.6 Proposed method versus other models on new dataset

The Swin-HSSAM model was compared with the AlexNet, ResNet 50, VGG16, MobileNet V2, ViT, and Cross ViT models on a public dataset (sourced from Kaggle at https://www.kaggle.com/datasets/gpiosenka/coffee-bean-dataset-resized-224-x-224), and the results are presented in Table 9. The Swin-HSSAM model achieved the highest scores, with a mAP of 97.83%, F1 score of 96.48%, and average accuracy of 96.50%, making it the top-performing model in the dataset. Comparative tests on the public dataset demonstrate the model's superior generalization capabilities.

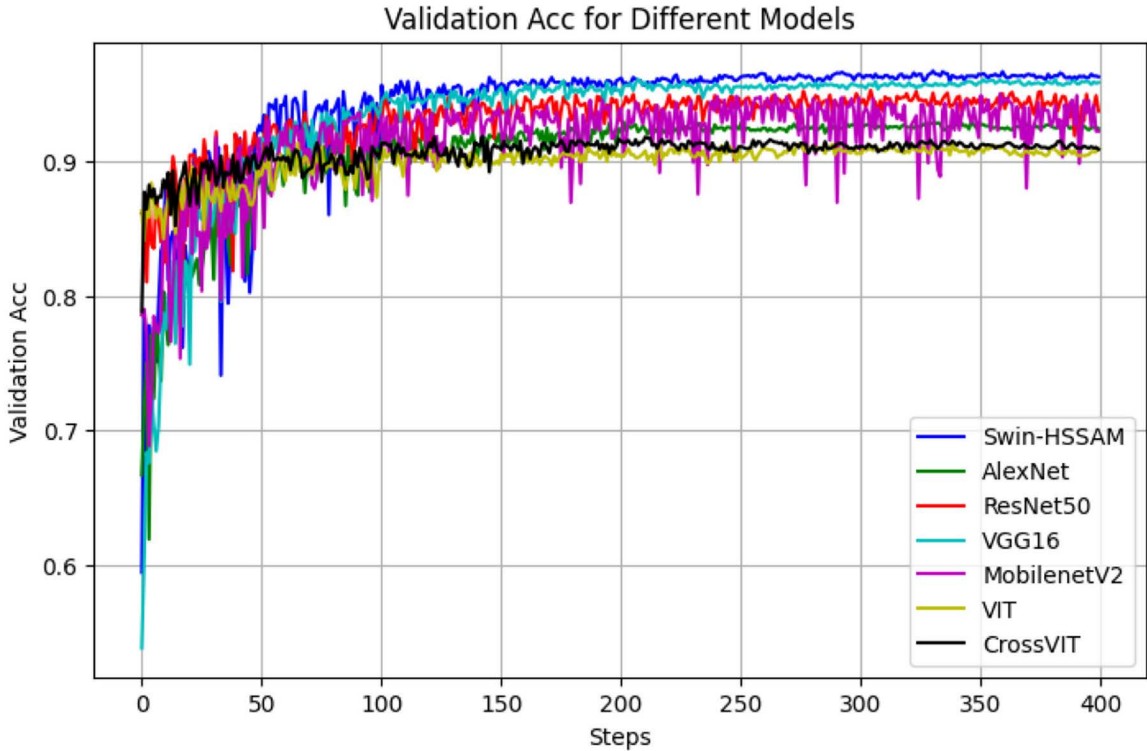

**Fig 15. Accuracy change curve of different mode.**

**Table 8. Comparison of complexity between different models.**

| Model | mAP | Average Accuracy (%) | F1 (%) | | | | Speed (FPS) | Params (M) | TheoFLOPs (B) |
|---|---|---|---|---|---|---|---|---|---|
| | | | First Grade | Second Grade | Third Grade | Defective | | | |
| Alexnet | 96.81 | 92.48 | 94.63 | 89.91 | 94.17 | 95.98 | 387.79 | 58.32 | 1.13 |
| ResNet50 | 98.04 | 93.78 | 95.84 | 93.02 | 94.74 | 98.05 | 133.29 | 25.63 | 16.16 |
| VGG16 | 97.93 | 95.90 | 97.58 | 91.94 | 95.95 | 97.38 | 318.26 | 138.36 | 15.47 |
| MobileNet V2 | 97.59 | 92.29 | 95.39 | 90.98 | 94.74 | 97.20 | 108.44 | 3.51 | 2.43 |
| VIT | 96.20 | 90.98 | 96.34 | 88.39 | 92.37 | 95.26 | 114.30 | 85.5 | 17.61 |
| CrossViT | 96.63 | 90.94 | 95.56 | 90.95 | 92.94 | 95.00 | 110.32 | 44.21 | 9.50 |
| **Swin-HSSAM** | **98.51** | **96.34** | **97.63** | **93.52** | **96.56** | **97.50** | **114.15** | **28.75** | **4.37** |

mAP, mean average precision.

## 4. Discussion

The above results demonstrate that the Swin-T network exhibited high accuracy in small target detection, and the fusion of the HS-FPN and SAM module were highly efficient in extracting features. This enhanced performance is suitable for multiple types of small target detection tasks. To reduce the loss while addressing the problem of hard-to-distinguish samples (e.g., sour versus immature beans in defective beans), we incorporated Fusion Loss again and ultimately proposed the Swin-HSSAM model.

**Table 9. Comparison of complexity between different models on new dataset.**

| Model | mAP | Average Accuracy (%) | F1 (%) |
|---|---|---|---|
| AlexNet | 90.46 | 95.49 | 95.49 |
| ResNet50 | 88.71 | 96.37 | 96.36 |
| VGG16 | 80.90 | 91.75 | 91.81 |
| MobileNetV2 | 91.80 | 95.65 | 95.44 |
| VIT | 93.44 | 95.77 | 95.78 |
| CrossViT | 93.93 | 96.23 | 96.23 |
| **Swin-HSSAM** | **97.83** | **96.50** | **96.48** |

mAP, mean average precision.

The Swin-HSSAM model is superior to the mainstream target detection models AlexNet, VGG16, ResNet50, MobileNet-v2, Vision Transformer (ViT), and CrossViT. It incorporates the HS-FPN and SAM modules, and the simultaneous.,and Introducing Fusion Loss considerably improved model accuracy and recall, implying that the model fully extracts the rich features of raw coffee beans and effectively labels and classifies raw coffee beans into different categories. Complex situations exist in raw coffee beans. First, the size difference between normal beans of three classes is minimal. In particular, the size difference between second- and third-class beans is even smaller. Second, the features of different defective beans are diverse. Differences exist in the detection performance of mainstream target detection models on each category; however, the detection performance of the Swin-HSSAM model on each category is higher than those of other mainstream target detection models; this demonstrates that incorporating the HS-FPN, SAM module, and Fusion Loss enables the accurate capture of detailed information on different scales, enriches the linguistic information of features, and improves the accuracy and comprehensiveness of image understanding. Notably, the Swin-HSSAM model has a higher mAP and a considerably improved average accuracy than ResNet. Furthermore, the model can obtain relatively high F1 values in the classification of first-, second-, and third-grade raw beans.

The accurate identification of different defective beans is challenging in raw coffee bean classification because the characteristics of different defective coffee beans are complex. For example, parchment beans have shape and color similar to normal beans, while sour beans are similar to fungus-damaged beans in color. Thus, distinguishing the color and texture characteristics of black, sour, and moldy beans with various degrees of severity is difficult. The Swin-HSSAM model showed superiority in classifying nine types of defective beans. Table 10 illustrates the classification results, implying that the mAP of black, broken, cherry pods, husk, shells, and sour were >95%. The mAP of fungus damage and immature were >90%, while the mAP of parchment was 88.86%, which was the only mAP lower than 90%. These results demonstrate the advantage of the Swin-HSSAM model in extracting feature information (color, edge, and texture).

**Table 10. Comparison of complexities between different models based on nine defects.**

| Model | mAP | | | | | | | | |
|---|---|---|---|---|---|---|---|---|---|
| | **Black** | **Broken** | **Cherry pods** | **Fungus damage** | **Husk** | **Immature** | **Parchment** | **Shells** | **Sour** |
| Alexnet | **98.35** | **83.44** | **97.34** | **83.12** | **97.08** | **95.53** | **95.44** | **94.58** | **95.29** |
| ResNet50 | **98.93** | **98.18** | **97.23** | **72.82** | **92.75** | **95.10** | **99.98** | **96.96** | **68.73** |
| VGG16 | **98.44** | **94.57** | **97.23** | **89.22** | **95.23** | **97.23** | **95.22** | **87.20** | **84.22** |
| MobileNet V2 | **98.00** | **97.58** | **97.21** | **90.42** | **95.74** | **95.44** | **94.21** | **98.55** | **97.42** |
| VIT | **98.47** | **98.14** | **98.55** | **84.54** | **99.21** | **94.33** | **93.44** | **87.54** | **88.22** |
| CrossViT | **96.00** | **92.50** | **94.30** | **88.21** | **95.15** | **94.75** | **98.74** | **66.72** | **90.40** |
| **Swin-HSSAM** | **98.50** | **99.11** | **99.42** | **93.35** | **99.50** | **92.80** | **88.86** | **98.00** | **95.84** |

Comparisons with publicly datasets have shown that the superior generalization ability of the Swin-HSSAM model. The images in the public dataset differ from those in the self-constructed (proprietary) dataset concerning illumination, size, and resolution. Furthermore, the coffee beans in this dataset are roasted for classification. However, the Swin-HSSAM model still obtained higher mAP values, average accuracy, and F1 score. Thus, the Swin-HSSAM model has the ability for generalization, exhibits better stability against interference, and can adapt to a larger range of new and unknown situations.

The Swin-HSSAM model included a moderately high level number of parameters. Specifically, the Swin-HSSAM model had a higher number of parameters than the lightweight model MobileNet V2, slightly higher number than ResNet50 and a lower number than other models; this can be attributed to HS-FPN through the hierarchical structure and adaptive feature selection mechanism. The inference speed of the Swin-HSSAM model is slightly higher than those of speed than MobileNet V2 and CrossViT, comparable to that of VIT and lower than those of other comparison models, such as ResNet50. Notably, the inference speed of the Swin-HSSAM model was mainly affected by the model configurations and hardware devices because current GPU devices are especially optimized for Transformer class models. Furthermore, the use of some advanced techniques, such as parallel computing and hardware acceleration, can potentially further improve the inference speed of the model. With the availability of GPU devices specifically optimized for Transformer class models, the performance of our Swin-HSSAM model will be substantially improved, making it suitable for resources-constrained environments.

Future research should further explore the model's capacity to accurately identify a wider array of defects, ideally covering all defect categories enumerated by the SCAA association. Currently, the Swin-HSSAM model categorized green coffee beans into four classes; however, the grading of coffee beans in commercial transactions could be segmented into finer categories. Future studies could refine the model to meet the comprehensive grading requirements for all green coffee beans. Furthermore, further investigations should employ the Swin-HSSAM model to evaluate other fruits, such as cherries and tomatoes.

## 5. Conclusions

This paper described the development of the novel Swin-HSSAM method, which is based on the Swin Transformer, for grading green coffee beans. This innovative approach leverages the strengths of the Transformer network, SAM module, and HS-FPN module and is equipped with pretrained weights to extract image features. These features are then fused via HS-FPN and enhanced by the SAM before being inputted into a classification head to predict labels. During the experimental phase, the proposed Swin-HSSAM method generated impressive results compared with combinations of the Swin Transformer with other classifiers, achieving mAP, accuracy, and F1 scores of 98.51%, 96.34%, and 96.35%, respectively. Extensive experiments conducted on green coffee bean grading demonstrated the exceptional performance of the proposed Swin-HSSAM method. Notably, the Swin Transformer served as an efficient feature extractor. The proposed method was proven effective and demonstrates broad potential for use in identifying other fruits and vegetables, as well as promising applications in agricultural product sorting.

## Author contributions

**Conceptualization:** Yujie Jiao.

**Data curation:** Yujie Jiao, Tianyun Wang.

**Formal analysis:** Yujie Jiao, Aoying Jia, Tianyun Wang.

**Funding acquisition:** Yuqing Zhao.

**Investigation:** Yuqing Zhao, Kaiming Xiang.

**Methodology:** Yujie Jiao, Yuqing Zhao.

**Project administration:** Yujie Jiao, Yuqing Zhao.

**Resources:** Yujie Jiao.

**Software:** Yujie Jiao, Jiashun Li.

**Supervision:** Yuqing Zhao, Tianyun Wang, Hangyu Deng, Maochang He, Yue Zhang.

**Validation:** Rui Jiang, Yue Zhang.

**Visualization:** Yue Zhang.

**Writing – original draft:** Yujie Jiao.

**Writing – review & editing:** Yuqing Zhao.

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
