## [Decision Letter · Decision Letter 0]

29 Oct 2024

PONE-D-24-41028Swin-HSSAM: A green coffee bean grading method by Swin TransformerPLOS ONE

Dear Dr. Zhang,

Thank you for submitting your manuscript to PLOS ONE. After careful consideration, we feel that it has merit but does not fully meet PLOS ONE’s publication criteria as it currently stands. Therefore, we invite you to submit a revised version of the manuscript that addresses the points raised during the review process.

**ACADEMIC EDITOR: Please make sure that the manuscript follows the format of the journal. References must be properly cited before resubmitting the revised paper.**

We look forward to receiving your revised manuscript.

Kind regards,

Nattapol Aunsri, Ph.D.

Academic Editor

PLOS ONE

Journal Requirements:

3. Please note that PLOS ONE has spec6ific guidelines on code sharing for submissions in which author-generated code underpins the findings in the manuscript. In these cases, all author-generated code must be made available without restrictions upon publication of the work. Please review our guidelines at https://journals.plos.org/plosone/s/materials-and-software-sharing#loc-sharing-code and ensure that your code is shared in a way that follows best practice and facilitates reproducibility and reuse.

4. Funding Information and Financial Disclosure sections do not match:

We note that the grant information you provided in the ‘Funding Information’ and ‘Financial Disclosure’ sections do not match. 

5. Please provide a complete Data Availability Statement in the submission form, ensuring you include all necessary access information or a reason for why you are unable to make your data freely accessible. If your research concerns only data provided within your submission, please write "All data are in the manuscript and/or supporting information files" as your Data Availability Statement.

Reviewers' comments:

Reviewer's Responses to Questions

**Comments to the Author**

1. Is the manuscript technically sound, and do the data support the conclusions?

Reviewer #1: Yes

Reviewer #2: Yes

2. Has the statistical analysis been performed appropriately and rigorously? 

Reviewer #1: Yes

Reviewer #2: Yes

3. Have the authors made all data underlying the findings in their manuscript fully available?

Reviewer #1: Yes

Reviewer #2: No

4. Is the manuscript presented in an intelligible fashion and written in standard English?

Reviewer #1: Yes

Reviewer #2: Yes

5. Review Comments to the Author

Reviewer #1: "Swin-HSSAM: A Green Coffee Bean Grading Method by Swin Transformer" presents a deep learning-based model for accurate coffee bean classification. Swin-HSSAM combines a Swin Transformer backbone with feature fusion via HS-FPN and SAM modules, and employs Fusion Loss to enhance performance. It outperforms models like AlexNet, VGG16, and ResNet50, achieving a 96.34% accuracy on a proprietary dataset and also performs well on a public dataset. The model offers a promising solution for automating coffee bean grading and may extend to other agricultural products.

The shortcoming include:

1. Lack of broader data validation:

Although the paper uses a proprietary dataset and one public dataset for validation, the diversity and scale of the datasets are relatively limited.

2. Insufficient analysis of model complexity and computational cost:

While the Swin Transformer, along with the HS-FPN and SAM modules, improves performance, the computational complexity and training time of these deep learning models may be high.

3. Lack of detail in defect bean classification:

The paper mentions that the classification of defective beans in the grading process is somewhat vague, and while the authors introduce SCA-recommended categories, the classification of defective beans may still lack precision.

NEED provide source code and dataset on Github on Next version.

References:

S. Hao et al., "ST-Double-Net: A Two-Stage Breast Tumor Classification Model Based on Swin Transformer and Weakly Supervised Target Localization," in IEEE Access, vol. 12, pp. 117921-117933, 2024, doi: 10.1109/ACCESS.2024.3445954.

Reviewer #2: 1. The manuscript lacks a clear Data Availability Statement, as PLOS ONE's policy requires. While the authors mention using a proprietary dataset and a publicly available dataset from Kaggle, it is unclear if the proprietary dataset is made fully available without restriction. The authors should explicitly provide access to the data or clarify any restrictions related to its availability. Additionally, the dataset link provided seems to be incorrect or broken and should be reviewed.

2. The use of acronyms (e.g., mAP, SAM, HS-FPN) is consistent, but it would be helpful to define these terms more clearly early on for readers unfamiliar with the specific methods.

3. The authors should present a more comprehensive analysis in discussion part.

4. Before submitting your manuscript, I strongly recommend reviewing the formatting to align with PLOS ONE’s submission guidelines. Several formatting issues, such as missing references (e.g., "Error! Reference source not found"), should be corrected. These errors can significantly impact the professionalism and readability of the manuscript. Please refer to the journal's submission guidelines to ensure all requirements are met.

6. PLOS authors have the option to publish the peer review history of their article (what does this mean? ). If published, this will include your full peer review and any attached files.

**Do you want your identity to be public for this peer review?** For information about this choice, including consent withdrawal, please see our Privacy Policy .

Reviewer #1: No

Reviewer #2: No

---

## [Author Response · Author response to Decision Letter 1]

12 Dec 2024

Response to Reviewers

Reviewer #1:

1. Lack of broader data validation:

Although the paper uses a proprietary dataset and one public dataset for validation, the diversity and scale of the datasets are relatively limited.

Response We have searched for a large number of datasets on the web, but the datasets that meet the requirements for the topic of our project on raw coffee bean identification are quite sparse, and the public and proprietary datasets used in the current article are already the larger datasets currently available, and we have asked relevant articles in the IEEE access journals about the possibility of sharing the datasets, but there has been no response.

2. Insufficient analysis of model complexity and computational cost:

While the Swin Transformer, along with the HS-FPN and SAM modules, improves performance, the computational complexity and training time of these deep learning models may be high.

Response Comparison and analysis of model params and theo flops have been added to the experimental section, and corresponding parameter data have been added to the tables.

3. Lack of detail in defect bean classification:

The paper mentions that the classification of defective beans in the grading process is somewhat vague, and while the authors introduce SCA-recommended categories, the classification of defective beans may still lack precision.

Response Details of defect categories and corresponding map accuracy and F1 values for each defect type have been added, tables and analyses have been added to the experimental section.

Reviewer #2:

4. The manuscript lacks a clear Data Availability Statement, as PLOS ONE's policy requires. While the authors mention using a proprietary dataset and a publicly available dataset from Kaggle, it is unclear if the proprietary dataset is made fully available without restriction. The authors should explicitly provide access to the data or clarify any restrictions related to its availability. Additionally, the dataset link provided seems to be incorrect or broken and should be reviewed.

Response Proof of data availability has been added, links to the dataset have been fixed, and the specified pages can now be viewed. The shared and proprietary datasets used in the text are available and the code and some of the datasets have been shared to the web page (https://github.com/tonyalice77/Papers-related-to-swin-transformer-improvements/tree/master) The original files of the datasets are quite large and will be uploaded later, if you have any other questions, please contact us with 690822961@qq.com

5. The use of acronyms (e.g., mAP, SAM, HS-FPN) is consistent, but it would be helpful to define these terms more clearly early on for readers unfamiliar with the specific methods.

Response Notes have been added for each term where it first appears to help readers better understand the text

6. The authors should present a more comprehensive analysis in discussion part.

Response The discussion has been supplemented with a more detailed analysis of model performance, types of defective beans, and model interpretability, as well as an outlook on what needs to be added to the model and what can be improved afterward

7. Before submitting your manuscript, I strongly recommend reviewing the formatting to align with PLOS ONE’s submission guidelines. Several formatting issues, such as missing references (e.g., "Error! Reference source not found"), should be corrected. These errors can significantly impact the professionalism and readability of the manuscript. Please refer to the journal's submission guidelines to ensure all requirements are met.

Response Changes have been made to the manuscript in accordance with the PLOS submission guidelines, such as changing the citation format.

---

## [Editor Report · Decision Letter 1]

16 Dec 2024

PONE-D-24-41028R1Swin-HSSAM: A green coffee bean grading method by Swin TransformerPLOS ONE

Dear Dr. Zhang,

Thank you for submitting your manuscript to PLOS ONE. After careful consideration, we feel that it has merit but does not fully meet PLOS ONE’s publication criteria as it currently stands. Therefore, we invite you to submit a revised version of the manuscript that addresses the points raised during the review process.

**ACADEMIC EDITOR: The paper still has Errors in References, The messages "Error! Reference source not found" are all over the manuscript. I will not pass your revised paper to the reviewers unless these errors have been fixed.**

We look forward to receiving your revised manuscript.

Kind regards,

Nattapol Aunsri, Ph.D.

Academic Editor

PLOS ONE

---

## [Author Response · Author response to Decision Letter 2]

17 Dec 2024

The citations have been reworked, and DOI numbers have been added to all but some of the ancient literature, of which the Chinese literature has also been specifically noted and given a DOI number.

---

## [Decision Letter · Decision Letter 2]

22 Jan 2025

PONE-D-24-41028R2Swin-HSSAM: A green coffee bean grading method by Swin TransformerPLOS ONE

Dear Dr. Zhang,

Thank you for submitting your manuscript to PLOS ONE. After careful consideration, we feel that it has merit but does not fully meet PLOS ONE’s publication criteria as it currently stands. Therefore, we invite you to submit a revised version of the manuscript that addresses the points raised during the review process.

**ACADEMIC EDITOR: Please clearly define "Author contributions" of you paper. There are many more authors in the revised manuscript than the original one. Please clearly state the acceptable reasons as well. In addition, please also fix dataset availability of the paper.**

We look forward to receiving your revised manuscript.

Kind regards,

Nattapol Aunsri, Ph.D.

Academic Editor

PLOS ONE

Journal Requirements:

Reviewers' comments:

Reviewer's Responses to Questions

**Comments to the Author**

1. If the authors have adequately addressed your comments raised in a previous round of review and you feel that this manuscript is now acceptable for publication, you may indicate that here to bypass the “Comments to the Author” section, enter your conflict of interest statement in the “Confidential to Editor” section, and submit your "Accept" recommendation.

Reviewer #1: All comments have been addressed

Reviewer #2: All comments have been addressed

2. Is the manuscript technically sound, and do the data support the conclusions?

Reviewer #1: Yes

Reviewer #2: Yes

3. Has the statistical analysis been performed appropriately and rigorously? 

Reviewer #1: Yes

Reviewer #2: Yes

4. Have the authors made all data underlying the findings in their manuscript fully available?

Reviewer #1: Yes

Reviewer #2: Yes

5. Is the manuscript presented in an intelligible fashion and written in standard English?

Reviewer #1: Yes

Reviewer #2: Yes

6. Review Comments to the Author

Reviewer #1: The paper title "Swin-HSSAM: A Green Coffee Bean Grading Method by Swin Transformer" has undergone careful revisions and is ready for acceptance. but lost two references.

For ResNet

E. Jing, H. Zhang, Z. Li, Y. Liu, Z. Ji, and I. Ganchev, "ECG heartbeat classification based on an improved ResNet-18 model," Computational and Mathematical Methods in Medicine, vol. 2021, 2021.

For Swin Transformer

S. Hao et al., "ConvNeXt-ST-AFF: A Novel Skin Disease Classification Model Based on Fusion of ConvNeXt and Swin Transformer," IEEE Access, 2023.

Reviewer #2: Thank you for submitting your revised manuscript

I appreciate the effort you’ve put into addressing the feedback so far. However, I noticed that the proprietary dataset is still pending availability, which could make it difficult for others to reproduce your findings.

I also noticed that the number of authors has significantly increased since the initial submission. Could you clarify the specific contributions of the newly added authors to ensure transparency?

7. PLOS authors have the option to publish the peer review history of their article (what does this mean? ). If published, this will include your full peer review and any attached files.

**Do you want your identity to be public for this peer review?** For information about this choice, including consent withdrawal, please see our Privacy Policy .

Reviewer #1: No

Reviewer #2: No

---

## [Author Response · Author response to Decision Letter 3]

25 Feb 2025

Reviewer #1:

1. The paper title "Swin-HSSAM: A Green Coffee Bean Grading Method by Swin Transformer" has undergone careful revisions and is ready for acceptance. but lost two references.

For ResNet

E. Jing, H. Zhang, Z. Li, Y. Liu, Z. Ji, and I. Ganchev, "ECG heartbeat classification based on an improved ResNet-18 model," Computational and Mathematical Methods in Medicine, vol. 2021, 2021.

For Swin Transformer

S. Hao et al., "ConvNeXt-ST-AFF: A Novel Skin Disease Classification Model Based on Fusion of ConvNeXt and Swin Transformer," IEEE Access, 2023.

Response We have already added these two references into the citation list.

Reviewer #2:

2.I appreciate the effort you’ve put into addressing the feedback so far. However, I noticed that the proprietary dataset is still pending availability, which could make it difficult for others to reproduce your findings.

I also noticed that the number of authors has significantly increased since the initial submission. Could you clarify the specific contributions of the newly added authors to ensure transparency?

Response After selecting 'accept' in the document, the dataset was made public. Therefore, we will complete the dataset once the acceptance is confirmed.

The reason for adding authors is that our manuscript was not as detailed during the initial submission, hence only a few principal authors were included. It was always our intention to add these additional authors in subsequent revisions. The specific contributions of these authors are as follows:

Hangyu Deng help assisted in resolving certain software issues during the image processing phase.

Kaiming Xiang contributed to the investigative research on the grading standards of green coffee beans and the actual conditions of green coffee beans.

Maochang He assisted in overseeing the rationality of the experiments and helped verify whether my tests served a cross-validating purpose.

Rui Jiang helped verify whether my tests served a cross-validating purpose.

Aoying Jia assisted in the analysis of data and the correction of errors in data comparison.

Jiashun Li He was responsible for the proofreading and correction of the main program, and he assisted me in fixing bugs when they occurred.

---

## [Decision Letter · Decision Letter 3]

18 Mar 2025

Swin-HSSAM: A green coffee bean grading method by Swin Transformer

PONE-D-24-41028R3

Dear Dr. Zhang,

We’re pleased to inform you that your manuscript has been judged scientifically suitable for publication and will be formally accepted for publication once it meets all outstanding technical requirements.

Kind regards,

Nattapol Aunsri, Ph.D.

Academic Editor

PLOS ONE

Additional Editor Comments (optional):

Reviewers' comments:

Reviewer's Responses to Questions

**Comments to the Author**

1. If the authors have adequately addressed your comments raised in a previous round of review and you feel that this manuscript is now acceptable for publication, you may indicate that here to bypass the “Comments to the Author” section, enter your conflict of interest statement in the “Confidential to Editor” section, and submit your "Accept" recommendation.

Reviewer #2: All comments have been addressed

2. Is the manuscript technically sound, and do the data support the conclusions?

Reviewer #2: Yes

3. Has the statistical analysis been performed appropriately and rigorously? 

Reviewer #2: Yes

4. Have the authors made all data underlying the findings in their manuscript fully available?

Reviewer #2: Yes

5. Is the manuscript presented in an intelligible fashion and written in standard English?

Reviewer #2: Yes

6. Review Comments to the Author

Reviewer #2: Thank you for your responses .I see that the dataset link is now accessible with sample data. I trust that the full dataset will be made available in the final acceptance stage. To ensure transparency, please make sure all necessary data is uploaded and clearly referenced in the manuscript.

7. PLOS authors have the option to publish the peer review history of their article (what does this mean? ). If published, this will include your full peer review and any attached files.

**Do you want your identity to be public for this peer review?** For information about this choice, including consent withdrawal, please see our Privacy Policy .

Reviewer #2: No

---

## [Editor Report · Acceptance letter]

PONE-D-24-41028R3

PLOS ONE

Dear Dr. Zhang,

I'm pleased to inform you that your manuscript has been deemed suitable for publication in PLOS ONE. Congratulations! Your manuscript is now being handed over to our production team.

Kind regards,

on behalf of

Dr. Nattapol Aunsri

Academic Editor

PLOS ONE